# Synovial Concentration of Trimethoprim-Sulphadiazine Following Regional Limb Perfusion in Standing Horses

**DOI:** 10.3390/ani11072085

**Published:** 2021-07-13

**Authors:** Kajsa Gustafsson, Amos J. Tatz, Roee Dahan, Wiessam Abu Ahmad, Malka Britzi, Gila A. Sutton, Gal Kelmer

**Affiliations:** 1Department of Large Animal Medicine and Surgery, Koret School of Veterinary Medicine, The Robert H. Smith Faculty of Agriculture, Food and Environment, The Hebrew University of Jerusalem, Rehovot 76100, Israel; kajsagustafsson1@gmail.com (K.G.); tatz@agri.huji.ac.il (A.J.T.); roee.dahan@mail.huji.ac.il (R.D.); gila.sutton@mail.huji.ac.il (G.A.S.); 2Faculty of Medicine, Hebrew University-Hadassah Braun School of Public Health and Community Medicine, Jerusalem 9112102, Israel; wiessam@gmail.com; 3National Residue Control Laboratory, Kimron Veterinary Institute, Bet Dagan 50250, Israel; malkab@moag.gov.il

**Keywords:** horse, pharmacokinetics, TMS, regional limb perfusion

## Abstract

**Simple Summary:**

Trimethoprim-sulphadiazine is a first line antimicrobial drug recommended for use in equine orthopedic infections such as deep wounds, ulcerative lymphangitis and septic arthritis. The pharmacokinetics of trimethoprim-sulphadiazine, when delivered through intravenous regional limb perfusion, has not been previously described. This study aimed at describing the pharmacokinetics and safety of the administration of trimethoprim-sulphadiazine through a single cephalic vein injection. Several horses in the study suffered from severe vasculitis, and the resulting synovial fluid concentration of trimethoprim-sulphadiazine over time was low. In light of these findings, the administration of trimethoprim-sulphadiazine to horses using regional limb perfusion via cephalic appears unwarranted.

**Abstract:**

The aim of this study was to investigate the safety and pharmacokinetics of trimethoprim-sulphadiazine administered via intravenous regional limb perfusion (IVRLP) into the cephalic vein. According to the hypothesis, the drug could be administered without adverse effects and the synovial concentrations would remain above the minimum inhibitory concentration (MIC) for trimethoprim-sulphadiazine (0.5 and 9.5 µg/mL) for 24 h. Ten (*n* = 10) horses underwent cephalic vein IVRLP with an Esmarch tourniquet applied for 30 min. Four grams (4 g) of trimethoprim-sulphadiazine (TMP-SDZ) were diluted at 0.9% NaCl for a total volume of 100 mL. Synovial fluid and blood samples were obtained immediately before IVRLP and at 0.25, 0.5, 2, 6, 12 and 24 h after the initiation of IVRLP. Trimethoprim and sulphadiazine concentrations were determined using a method based on liquid chromatography/tandem mass spectrometry. The C_max_ (peak drug concentration) values were 36 ± 31.1 and 275.3 ± 214.4 µg/mL (TMP and SDZ). The respective t_max_ (time to reach C_max_) values were 20 ± 7.8 and 26.4 ± 7.2 min. The initial synovial fluid concentrations were high but decreased quickly. No horse had synovial concentrations of trimethoprim-sulphadiazine above the MIC at 12 h. Severe vasculitis and pain shortly after IVRLP, lasting up to one week post-injection, occurred in five out of 10 horses. In conclusion, IVRLP with trimethoprim-sulphadiazine cannot be recommended due to the low concentrations of synovial fluid over time and the frequent severe adverse effects causing pain and discomfort in treated horses. Thus, in cases of septic synovitis with bacteria sensitive to trimethoprim-sulphadiazine, other routes of administration should be considered.

## 1. Introduction

Septic arthritis is a severe and potentially life-threatening infection of synovial structures in horses that has a poor prognosis regarding the return to athletic use or even survival if not treated successfully rapidly [1,2,3,4].

Infection of synovial structures of the distal limb can be caused by trauma, iatrogenic intervention (e.g., intra-articular injection or arthroscopic surgery) or via haematogenous spread [5]. The recommended treatment includes an immediate lavage and debridement of the affected synovial structure, preferably under endoscopic guidance, as well as appropriate aggressive antimicrobial therapy [2]. Successful treatment is likely to be dependent on the achievement of high concentrations of antimicrobials within the affected synovial structure. Favorable results are seen when local delivery techniques to the affected synovial structure are used, such as intravenous regional limb perfusion (IVRLP), or antibiotic-impregnated polymethyl methacrylate or plaster of Paris beads [6].

The use of intravenous or intraosseous routes for RLP is well described and provides high concentrations of selected antimicrobials to synovial structures, without the adverse effects associated with parental administration, such as colitis or nephrotoxicity [7,8,9,10].

Trimethoprim-sulphadiazine combination is a broad-spectrum antimicrobial drug that interrupts the folate metabolism of bacterial cells. The two antimicrobial drugs (trimethoprim and sulphadiazine) exhibit a synergism in inhibiting bacterial DNA synthesis in two stages, reducing the potential for antimicrobial resistance development to a greater extent than when either drug is used alone [11].

Although individually both drugs are bacteriostatic, when combined, trimethoprim-sulphadiazine has a broad-spectrum bactericidal antimicrobial activity against many Gram-positive and Gram-negative bacteria, although *Pseudomonas* spp., *Mycoplasma* spp. and a number of isolates of *Klebsiella* spp. are resistant [12]. Furthermore, the antimicrobial activity against anaerobic bacteria is poor despite culture and sensitivity tests showing the contrary, most probably due to high levels of folate present in sites of anaerobic infection [12].

With emerging antibiotic resistance in both human and veterinary medicine, an appropriate use of antibiotics and avoidance of the use of some specific antibiotics is advised. Trimethoprim-sulphadiazine is a first-line drug advised for use in orthopedic cases such as deep wounds and septic arthritis, but there are no described methods of local delivery techniques [13]. Additionally, there have been reports of multiresistant *Staphylococcus aureus* susceptible to trimethoprim-sulphadiazine. In these cases, specifically aimed antimicrobial treatment can be lifesaving [14].

The pharmacokinetics of trimethoprim-sulphadiazine administered by parenteral route have been well described in plasma, urine, synovial fluid, peritoneal fluid, endometrium and pulmonary epithelial lining in horses [15,16,17,18]. The established Minimal Inhibitory Concentration (MIC) of trimethoprim and sulphadiazine for the most commonly isolated bacteria in orthopedic infections is 0.5 µg/mL and 9.5 µg/mL, respectively [19].

The objectives of this study were to evaluate the safety and the pharmacokinetics of trimethoprim-sulphadiazine when administered via IVRLP to horses. Our hypothesis was that the administration would be safe, with no side effects, and that antimicrobial concentrations would remain above the MIC for 24 h, enabling a once daily administration.

## 2. Study Design

### 2.1. Animals

Ten horses were enrolled to the study, two geldings and eight mares. Their ages ranged from two to 18 years (mean 10 years), and their weights ranged from 285 kg to 560 kg (mean 400 kg). All horses underwent a thorough clinical examination, complete blood count, as well as serum creatinine, urea and electrolyte analyses. No abnormalities were observed. Dynamic orthopaedic examinations were unremarkable, and the anatomy of the front limbs in all horses appeared normal on palpation and static flexion. The horses were housed individually and had free access to water and hay throughout the study. Approval for the study was granted by the IACUC (Institutional Animal Care and Use Committee) of the university (MD-16-14812-3).

### 2.2. Regional Limb Perfusion and Sampling

Perfusion was performed in a similar fashion to previous studies from our laboratory [10]. One forelimb limb of each horse was randomly selected for IVRLP by coin toss. The distal portion of the forelimb was desensitized by anaesthetizing the ulnar, median and medial cutaneous antebrachial nerves with 10 mL of 2% mepivacaine HCl (Mepivacaine, Ceva Animal Health, Glenorie, Australia) per site. After that, hair over the cephalic vein at the level of the chestnut was clipped and prepared for aseptic venipuncture. Horses were sedated with detomidine hydrochloride (0.01 mg/kg, IV; Domosedan, Orion Pharma, Espoo, Finland) and butorphanol tartrate (0.01 mg/kg, IV; Butomidor, Richter Pharma AG, Wels, Austria). Two rolled gauzes (Sion Medical, Sderot, Israel), each being 10 cm long, were placed over the vasculature proximal to the intended site of catheterization. An 8-cm wide rubber tourniquet (Degania Medical, Degania Bet, Israel) was applied forcefully over the gauze rolls and around the limb to occlude blood flow.

A 23-gauge, 2-cm long butterfly catheter (Scalp Vein Set, Zhejiang Kindly Medical Devices Co Ltd., Binhai, China) was inserted in an aseptic manner into the cephalic vein about 5 cm distal to the tourniquet and aimed distally. The catheter was adhered to the skin with cyanoacrylate glue. A perfusate composed of 16 mL of trimethoprim-sulphadiazine (Trimethoprim (4%) Sulphadiazine (20%), Norbrook Laboratories, Northern Ireland) in 84 mL of sterile isotonic saline solution (Sodium Chloride, Braun, Melsungen, Germany), giving a total volume of 100 mL, was infused into the cephalic vein over a period of 2 min. The tourniquet was removed 30 min after the perfusate was injected. Horses were observed closely for abnormalities associated with the injection, such as inadvertent perivascular administration, swelling or signs of pain. The movements of the horse’s legs, other than weight shifting, were recorded during the procedure.

### 2.3. Synovial Fluid Sampling

Sampling was performed similarly to previous studies from our laboratory [8]. After preparing the skin of the lateral aspect of the joint for aseptic arthrocentesis, synovial fluid (1 mL) was collected from the lateral aspect of the metacarpophalangeal joint (MCPJ), with the limb in light flexion. A 21-gauge, 3.81-cm needle was inserted into a slight depression between the proximal phalanx, the proximal sesamoid bone and the third metacarpal bone, through the collateral sesamoidean ligament, until the joint was entered. Synovial fluid (1 mL) was aspirated into a 3-mL syringe. Synovial fluid was collected after the tourniquet was applied but before the perfusate was injected (T = 0) and at 0.25 h and 0.5 h after injection. The tourniquet was removed after the collection of the 30-min sample. Synovial fluid was collected from the same joint at 2, 6, 12 and 24 h post perfusion. One-gram amikacin sulphate (Amikacin, Anfarm Hellas, Attika, Greece) was injected into the MCPJ following the final synovial fluid sample collection because of the perceived risk of synovial infection caused by repeated arthrocentesis of the joint. The horses were sedated with xylazine HCl (Hydrogen Chloride) (0.5 mg/kg, IV; AnaSed, Akorn Animal Health, Vernon Hills, IL, USA) as needed, to prevent discomfort and movement during arthrocentesis. Using a 21-gauge, 3.81-cm needle, blood (2 mL) was collected aseptically from the jugular vein at the same time points at which the synovial fluid was harvested.

All samples of synovial fluid and blood were placed into a plastic sample tube containing no anticoagulants. After coagulation, the samples were centrifuged for 5 min at 1000 g, and the supernatant was collected and frozen at −80 °C. The limb was maintained in a bandage between each collection of synovial fluid and for two days after the final arthrocentesis. All horses received phenylbutazone (4.4 mg/kg, IV) after the last sampling and for the next two days (2.2 mg/kg, PO, q12 h). Horses were observed once daily for two weeks for lameness and swelling at the injection sites associated with venipuncture, IVRLP or arthrocentesis.

### 2.4. Determination of Antimicrobial Concentration

Samples of serum and synovial fluid (20 µL) were vortex-mixed with 0.5 mL bi-distilled water (Type 1, Micropure water purification system, Thermo Scientific, Asheville, NC, USA) and 0.5 mL acetonitrile (J.T. Baker, Deventeer, The Netherlands). After 5 min of centrifugation at 14,000 rpm, at 4 °C, an aliquot of supernatant was used for LC/MS/MS analysis. Calibration curves of suphadiazine (A2S, Saint Jean d’Illac, France) and trimethoprim (Dr. Ehrenstorfer, Germany) were prepared in serum and synovial fluid free from the presence of these compounds, within a concentration range of 0.25–50 µg/mL in synovial fluid and 0.1–5 µg/mL in serum. Samples exceeding the highest level of calibration were diluted and reanalyzed. Accuracy and precision were determined on quality control (QC) samples fortified with 0.25, 5 and 50 µg/mL in synovial fluid and 0.1, 1 and 5 µg/mL in serum. These values represent low-level (LL), medium-level (ML) and high-level (HL) calibration curves.

Sulphadiazine and trimethoprim were detected and quantified by liquid chromatography tandem mass spectrometry (LC/MS/MS). The system included an Agilent 1200 (Agilent Technologies, Waldbronn, Germany) liquid chromatography system (with a degasser, binary pump, temperature-controlled column compartment and autosampler) connected to an Applied Biosystems API 4000 mass spectrometer (Applied Biosystems, Toronto, ON, Canada).

Compounds were separated on a Zorbax, Eclipse Plus C18 column (1.8 μm, 4.6 × 50 mm, Agilent, Santa Clara, CA, USA). The mobile phase consisted of 0.1% formic acid (Sigma, St. Louis, MO, USA) and acetonitrile. The precursor ions were 251 and 291 m/z and the quantifier product ions were 156 and 230 m/z for sulphadiazine and trimethoprim, respectively.

### 2.5. Statistical Methods

WinPepi; Pairsetc. (Version 3.59); Power calculation P3 was used to calculate the power of the study. The data were assumed not to have a normal distribution based on the histogram and the small sample size. The levels of each antibiotic drug at the six different time points at each of the two locations were compared individually by the nonparametric Friedman Test for a repeated-measures 2-way Analysis of Variance (ANOVA) by ranks. *p*-values for the pair-wise comparisons were adjusted for multiple comparisons by the Bonferroni correction. Statistical significance was set at *p* < 0.05. Statistical analysis was carried out in IBM SPSS Version 25.

## 3. Results

During the course of the experimental protocol, five horses developed severe phlebitis of the cephalic vein, with associated cellulitis, which manifested in severe swelling, heat, pain upon palpation of the distal limb and lameness at walking. In one horse, pain and swelling of the distal limb made it inappropriate to perform any further arthrocentesis from 6 h post-infusion. Nevertheless, data obtained from this horse was included in the study. For the horses with phlebitis, local cryotherapy with ice bandages was applied for the first 24 h, and phenylbutazone (2.2 mg/kg, PO, q12h) administration was continued until the acute phlebitis resolved for 5–7 days, in addition to the standard two days. In three horses the lameness resolved within 1–2 days, and two horses showed lameness and reluctance to walk that lasted 3–5 days.

The power of the study was 72% for the sulphadiazine and 70% for the trimethoprim. Median (lower quartile; upper quartile) trimethoprim and sulphadiazine synovial C_max_ values were 19.2 (3.46; 50.6) and 175 (21.85; 394) µg ⁄mL. The respective t_max_ values were 15 (15; 30) and 15 (15; 15) minutes. The respective median serum C_max_ values were 3.5 and 65.6 µg ⁄mL, and the serum mean t_max_ values were 42 ± 43.8 and 114 ± 109.8 min.

The concentrations of trimethoprim and sulphadiazine in synovial fluid were significantly different over time, *p* = 0.031 and *p* = 0.021 respectively. The concentrations of trimethoprim and sulphadiazine in serum were not significantly different over time, *p* = 0.14 and *p* = 0.05 respectively.

Trimethoprim synovial fluid concentrations were initially high but decreased quickly, and only the concentrations at time points 0.25 and 0.5 h were statistically different from the synovial fluid concentrations at time point T0 (*p* < 0.01). Sulphadiazine synovial fluid concentrations remained elevated for longer, but nevertheless the synovial fluid concentrations at time points 0.25, 0.5 (*p* < 0.001) and 2 h (*p* < 0.01) were the only time points statistically different from the synovial fluid concentrations at time point T0. The trimethoprim and sulphadiazine synovial fluid concentration at 0.25 h was not statistically different from the synovial fluid concentration at time point 0.5 h (*p* = 1.00).

Since the data were not normally distributed, the synovial concentrations at each time point are described in medians with quartiles (Figure 1 and Figure 2).

The intraday accuracy of the synovial fluid QC samples was 91%, 99% and 94% for LL, ML and HL of sulphadiazine and 109%, 104% and 105% for trimethoprim. The precision (coefficient of variation) was 15%, 8.9% and 9.5% for LL, ML and HL of sulphadiazine and 2.1%, 5.1% and 5.4% for trimethoprim.

The interday accuracy of the serum QC samples was 107%, 97% and 94% for LL, ML and HL of sulphadiazine and 110%, 102% and 103% for trimethoprim. The precision (coefficient of variation) was 14%, 13.4% and 12.3% for LL, ML and HL of sulphadiazine and 14.8%, 7.1% and 4.4% for trimethoprim.

## 4. Discussion

This study aimed at investigating if trimethoprim-sulphadiazine administration to horses by IVRLP was safe and could maintain a synovial fluid concentration above MIC for 24 h.

The results revealed that trimethoprim-sulphadiazine administrated by IVRLP created severe and longstanding, apparently chemically induced, phlebitis in a high percentage of horses. The conclusion was that it cannot be administered safely to horses, at least with this formulation. In addition, the synovial fluid concentration of trimethoprim-sulphadiazine did not remain above MIC for the most common equine pathogens for 24 h, and both hypotheses have to therefore be rejected.

The horses that suffered from phlebitis were diagnosed with chemically induced phlebitis as it occurred in association with the intravenous infusion [20]. Chemically induced phlebitis is usually caused by injection of substances with high or low pH (<5 or >10) or substances with high osmolarity [21]. The trimethoprim-sulphadiazine formulation used in this study had a pH of 10, which could explain the reaction. Injection of substances with a substantially different pH to blood (7.35–7.45) causes inflammatory cell infiltration, edema, thrombosis and cell death via endothelial damage [21]. We could not find reports on vascular reaction to either the active ingredients (Trimethoprim and Sulphadiazine) or the main excipients (Chlorocresol and Sodium Formaldehyde Sulphoxylate Dihydrate). Clinical signs of phlebitis include tenderness, erythema, swelling and a palpable venous cord [22], which were all observed in the affected horses in this study. The decreased endothelial anticoagulant activity of the vessels exposed to irritating substances leads to thrombosis and loss of patency of the vessel. In these cases, thrombophlebitis was diagnosed [21]. Although clinically evident in the present study, the diagnosis of thrombophlebitis was not confirmed by further diagnostic tests such as ultrasonography.

The risk of chemically induced phlebitis can be decreased by diluting the injected substance. However, in this study, 16 mL of trimethoprim-sulphadiazine was diluted in 84 mL of saline for a total volume of 100 mL. Intravenous administration of trimethoprim-sulphadiazine to humans is prepared by diluting 5 mL in 75–125 mL of 5% dextrose, a higher dilution than was used in this study. Additionally, the 5% dextrose in sterile water is used as a diluent in humans because it mimics the osmolarity of blood and is thought to decrease the risk of a vascular reaction [20]. The phlebitis observed this study could potentially have been avoided by a higher dilution of the antimicrobial solution. However, a volume higher than 100 mL has not been investigated in IVRLP of the horse distal limb. Additionally, the administration time of trimethoprim-sulphadiazine in humans is recommended to be 60–90 min in order to obtain a satisfactory hemodilution during the infusion. In IVRLP, although the trimethoprim-sulphadiazine was diluted, the rapid injection time that was necessary, combined with the presence of a tourniquet when administering a drug through IVRLP, might not have provided sufficient hemodilution, resulting in the exposure of the tunica intima of the vein to high concentrations of the drug.

The dosage chosen in this study was based on the proposed dosage of 1/3 of the systemic dose when administering concentration-dependent antimicrobials through IVRLP [3]. The concentration-dependent activity of trimethoprim-sulphadiazine is debatable and although some concentration-dependent mode of action is evident, some authors also suggest a time-dependent activity of trimethoprim-sulphadiazine [23]. If a time-dependent mode of action were assumed, the dose would have to be increased to a full systemic dose, most likely resulting in even more detrimental effects on the vascular endothelium.

The low synovial fluid concentrations over time are particularly ineffective when using trimethoprim-sulphadiazine. Although the combination of trimethoprim-sulphadiazine is synergistic and can exert a bactericidal effect, both antimicrobial drugs are mainly bacteriostatic, and it is thus important to keep blood concentrations above MIC for as long a period of time as possible [15].

The volume of injection used in this study (100 mL) was based on previous studies concluding that a higher volume of infusion led to higher synovial fluid concentrations of the infused antimicrobial [24,25].

The synovial fluid concentration of trimethoprim-sulphadiazine was highly variable among the horses. The reason for the large variation could be related to several factors including blood contamination of synovial samples [26], transport of the perfusion to the systemic circulation through intraosseous vessels [27], or failure of the tourniquet caused by a faulty application or movement of the horse [28,29,30].

The use of Esmarch rubber tourniquets in this study was elected in order to mimic the most probable clinical setting. Although the use of pneumatic tourniquets would provide a more consistent protocol, the Esmarch rubber tourniquet has proved to be equally useful in this type of experimental protocol [31].

The fetlock was chosen as the sample site because in most other RLP PK studies that we performed [8,9,10,24,29] and in many other RLP PK studies [3,31], the fetlock was sampled. Conforming to a uniform and previously established study design is advantageous.

This study has several limitations including the small study group with highly variable synovial fluid trimethoprim-sulphadiazine concentrations. The sample size was based on previous studies where the sample size was large enough to find significant differences [24,25,28,29]. The same volume of antimicrobial was infused despite significant differences in body weight, and the antimicrobial concentrations were only measured in synovial fluid from the metacarpophalangeal joint. Thus, the antimicrobial concentration in other joints, skin, subcutaneous tissue and bone is unknown.

## 5. Conclusions

Based on the induction of severe and longstanding chemically induced phlebitis and a low synovial fluid concentration over time, trimethoprim-sulphadiazine could not be recommended for administration to horses via IVRLP.

## Figures and Tables

**Figure 1 animals-11-02085-f001:**
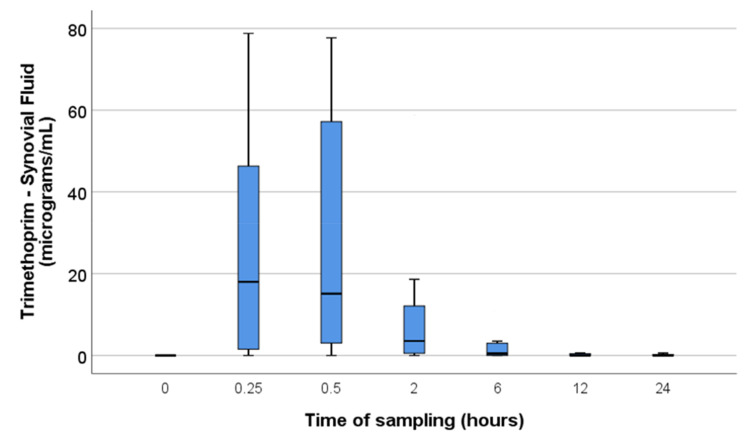
Trimethoprim synovial fluid concentrations at each time point. Median values and quartiles indicated by boxplots with maximum and minimum values indicated by whiskers. Synovial fluid concentrations were only significantly different from the concentrations at T0 at time points 0.25 and 0.5 h, although the median synovial fluid concentration remained above the MIC (0.5 µg/mL) until the 6-h time point.

**Figure 2 animals-11-02085-f002:**
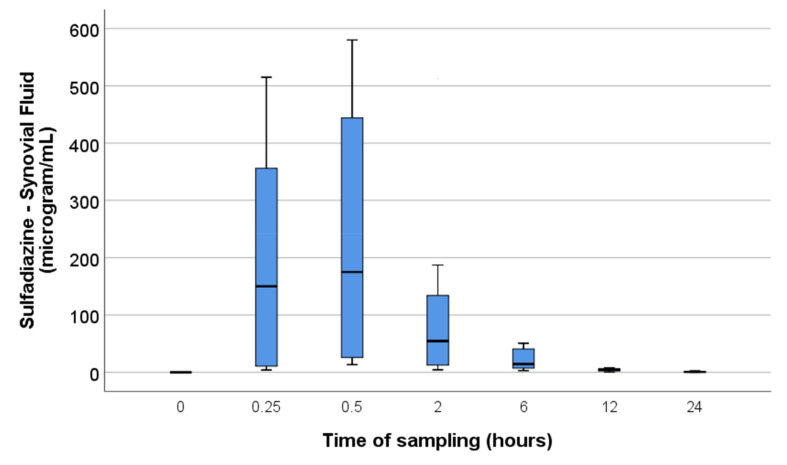
Sulphadiazine synovial fluid concentrations at each time point. Median values and quartiles indicated by boxplots with maximum and minimum values indicated by whiskers. The synovial fluid concentrations were only significantly different from the concentrations at T0 at time points 0.25, 0.5 and 2 h, although the median synovial fluid concentration remained above the MIC (9.5 µg/mL) until the 6-h time point.

## Data Availability

All original data is available upon request from the authors.

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
