# Peer review of "Synovial Concentration of Trimethoprim-Sulphadiazine Following Regional Limb Perfusion in Standing Horses"

_animals, 2021, doi:10.3390/ani11072085_

Round 1

Reviewer 1 Report

The article is well written. There are just couple of format changed on font size and type in the summary and lower in the manuscript. I marked them on the pdf file.

Couple questions came up while reading the study, that could be discussed.

First, why was only the fetlock joint tested and not also the carpus?

Second, TMPS has been reported to cause adverse reactions when injected IV after alpha-2 agonists medication, where you aware of this? was any adverse reaction noted? (out of the vasculitis?

It would be wise to discuss about the solution of the TMPS, so it is stated exactly what type of chemical is, and if it has been reported elsewhere as a cause of vascular reactions.

Reviewer 2 Report

This study describes serum and synovial fluid concentrations of trimethoprim and sulphadiazine (sulfadiazine) after administration by intravenous limb perfusion and concluded that at the dose used inadequate concentrations were achieved. Moreover 50% of treated horses had severe adverse reactions.

It is good to report negative outcomes.

However, in Europe trimethoprim and sulfadiazine would not be first line drugs for the treatment of an infected synovial cavity. I am therefore curious why the authors suggest that it might be. Does this reflect drug availability in Israel?  However. the authors used amikacin immediately after the study which would be a better first lime treatment. Further justification for the use of trimethoprim and sulfadiazine for treatment of acute joint infection is required.

American spelling of sulpahdiazine

No line numbers - annoying!

The study design is adequate, although the authors have not attempted to justify why they elected to use 10 horses. The Results could be better described. The Discussion is adequate.

Abstract

In the Abstract, Cmax & t max are not defined, nor is MIC.

'Treatment with trimethoprim-sulfadiazine in cases with septic synovitis should be administered conventionally by IV infusion.'

This sentence does not make sense as written - please rephrase

Introduction

Paragraph 2

antimicrobial drug therapy  

Better results than what? - & based on what evidence?

Why not intra-articular injection?

Would TMPS be a first line drug of choice? spectrum; bacteriostatic v bactericidal

Paragraph 5

You say in the Discussion that TMPS is mainly bacteriostatic. My understanding is that trimethoprim is bacteriostatic but, in some cases when combined with sulfadiazine may be bactericidal.

You need to be consistent.

Paragraph 7

MIC - define abbreviation

0.5μg/mL and 9.5 μg/mL, respectively

Study design       

2.1    IACUC    please write abbreviation in full      

giving a total volume of 100 mL

2.3 repeated arthrocentesis or multiple arthrocenteces

LC/MS/MS –Please define the abbreviations

2.5 The data were

            Results

Paragraph 1

When were these features first noted relative to the first injection?

Please make it clearer how many horses showed each reaction because here you imply 5 horses, whereas in the ultimate sentence in the paragraph you refer to 2 horses showing lameness.

What does 'pain of the distal limb' actually mean?

Please describe the distribution of the swelling and whether it was mild, moderate or severe.

Did the remaining 5 horses have absolutely no reaction? This would seem surprising if so.

Paragraph 2

Please give interquartile range and range in addition to the median so we have any idea of the spread of the data

Define Cmax  and tmax

Why have you presented a median value for  Cmax  and mean (& presumably standard deviation) for tmax?

Paragraph 3

P=0.031 and P=0.02, respectively.     P=0.14 and P=0.05, respectively.

Paragraph 4, line 2 0.5 hours

‘the synovial fluid concentrations at time points 0.25, 0.5 and 2 hours were the only time points statistically different from the synovial fluid concentrations at time point T0’ Please provide a p value.

concentrations  .... were...

Paragraph  5

data were

            Discussion

You inferred in the Introduction that the drug was bactericidal! Consistency & correctness are needed.

Authors' contributions to the study, funding, data availability statement are missing.

            References

Inconsistencies in journal citations  - some written in full, others abbreviated – need consistency
